# Hgt17-Adr1 Relationship in *Candida albicans* Citrate Utilization

**DOI:** 10.3390/jof11120889

**Published:** 2025-12-17

**Authors:** Amelia M. White, Aaron P. Mitchell

**Affiliations:** Department of Microbiology, University of Georgia, Athens, GA 30602, USA

**Keywords:** *Candida albicans*, carbon metabolism, transcriptional regulation

## Abstract

The fungal pathogen *Candida albicans* can infect diverse tissues, a reflection of its broad metabolic repertoire. The transcription factor Adr1 is required for utilization of several citric acid cycle intermediates that are found in tissue. Many Adr1-activated genes encode enzymes with well-defined roles in citrate metabolism or gluconeogenesis. Here, we focus on *HGT17* (C4_01070W, orf19.4682), an Adr1-activated gene that encodes a possible citrate transporter. We provide two lines of evidence that *HGT17* is a key functional target of Adr1. First, forced expression of *HGT17* in an *adr1*Δ/Δ mutant improves growth on citrate as a carbon source. Second, *hgt17*Δ/Δ and *adr1*Δ/Δ mutants incubated in citrate medium present similar gene expression defects compared to the wild type. Noteworthy is down-regulation in both mutants of citric acid cycle genes, glycolysis/gluconeogenesis genes, and ergosterol synthesis genes. These common features may reflect a specific effect of citrate as an inducer of citric acid cycle enzymes or a global effect of carbon and energy limitation. In either case, the results argue that reduced *HGT17* expression contributes substantially to the impact of an *adr1*Δ/Δ mutation on growth and gene expression.

## 1. Introduction

*Candida albicans* can survive and proliferate in a broad range of body sites as either a pathogen or a commensal [1,2,3,4]. Metabolic flexibility is critical for both pathogenic and commensal states [1,2,5,6]. This metabolic flexibility is coordinated by nutrient sensors and transcriptional regulatory networks [7,8].

Our focus here is the transcriptional control of citrate utilization. We recently found that the transcription factor Adr1, previously established to be a positive regulator of ergosterol biosynthetic genes [9], is required for growth on citrate and other citric acid cycle intermediates as sole sources of carbon [10]. RNA-sequencing (RNA-seq) data showed that many predicted citrate metabolic genes are downregulated in an *adr1*Δ/Δ mutant [10]. Adr1 may be part of a broad regulatory network that connects metabolism and virulence because epistasis analysis indicates that Adr1 may act downstream of virulence regulator Eed1 to govern citrate utilization [10,11].

One of the genes most dependent on Adr1 for expression is *HGT17* (C4_01070W, orf19.4682). It encodes a major facilitator superfamily member initially annotated as a glucose transporter [12]. Hgt17 lacks a precise ortholog in *Saccharomyces cerevisiae* [12], which cannot use citrate as a carbon source [13]. An *hgt17*Δ/Δ mutant is unable to grow on citrate but is capable of growth on other citric acid cycle intermediates [10]. We proposed that Hgt17 is a citrate transporter to account for these observations [10].

Given the strong dependence of *HGT17* expression on Adr1, we considered the possibility that reduced *HGT17* expression may be a pivotal determinant of *adr1*Δ/Δ mutant phenotypes in citrate growth medium. Here, we have taken two approaches to make an appraisal of that idea. Our findings indicate that *HGT17* expression has strong impact on the *adr1*Δ/Δ phenotype and suggest that citrate levels are critical for Adr1 function.

## 2. Materials and Methods

### 2.1. Media and Culture Conditions

Strains (Appendix A) were maintained in 15% glycerol stocks and stored at −80 °C. For use, strains were streaked on YPD agar plates (2% dextrose, 2% Bacto peptone, 1% yeast extract, 2% Bacto Agar [all from ThermoFisher, Waltham, MA, USA]). Single colonies were transferred to 18 h overnight cultures at 30 °C with agitation. Overnight cultures were grown in filter-sterilized 0.67% yeast nitrogen base without amino acids (ThermoFisher, Waltham, MA, USA) + 1% sodium acetate (VWR, Radnor, PA, USA) liquid medium. For the experiment using BPS in growth tests, overnight cultures included 50 µM BPS (Sigma-Aldrich, St. Louis, MO, USA), and plates included 50 or 100 µM BPS as indicated; BPS increases *RBT5* promoter activity [14].

Gene expression experiments were conducted with filter-sterilized YNB containing 1% sodium citrate (VWR, Radnor, PA, USA). All *C. albicans* transformants were selected as described [10] on Complete Supplement Mixture lacking histidine (CSM-HIS; 0.67% yeast nitrogen base without amino acids, 0.079% CSM-HIS supplement mixture [ThermoFisher, Waltham, MA, USA], and 2% dextrose) for His+ isolates or on YPD+NAT (2% Bacto peptone, 2% dextrose, 1% yeast extract, and 400 μg/mL nourseothricin [GoldBio, St. Louis, MO, USA]) for nourseothricin-resistant isolates.

### 2.2. Strain Construction

*C. albicans* genetic manipulations were conducted as previously described [10,15,16], starting with SC5314 derivative MC5 [17]. Plasmids [14,15,16,18,19] and primer sequences are listed in Appendix A. Mutants were constructed in the SC5314 strain background. The *HGT17* overexpression mutants were constructed in a background that had been made sensitive to nourseothricin by recycling the *NAT1* marker at the *his1*Δ/Δ locus simultaneously with the deletion of *ADR1* using *HIS1*, following the procedure of Huang and Mitchell [19]. The *HGT17* overexpression strains were then constructed by replacing the native promoter with the inducible *RBT5* promoter. The promoter cassette was amplified using primers with 80 bp of homology upstream or downstream to the native *HGT17* promoter region. The addition of adapter sequences to these 80 bp primers allowed the amplification of the *RBT5* promoter cassette with an attached *NAT* resistance marker from plasmid pTH10 [14]. Note that both *RBT5-HGT17* strains are heterozygous for the promoter replacement (genotype: *HGT17/NAT-RBT5-HGT17*).

The Nat-sensitive *adr1*Δ/Δ cells were transformed with 1 μg of CaCas9 DNA, 1 μg of an sgRNA targeted to the *HGT17* promoter region, and 3 μg of the *NAT1+RBT5* promoter cassette PCR product as repair template. Transformants were plated on YPD+NAT (2% Bacto peptone, 2% dextrose, 1% yeast extract, and 400 mg/mL nourseothricin) plates for 48 h. Candidates were genotyped by PCR using primers “HGT17 far up check /F” and “NAT1 check int /R” for the presence of the *NAT1*+*RBT5* promoter construct and primers “HGT17 check up /F” and “HGT17 check int /R” for the absence of the *HGT17* promoter region.

### 2.3. Agar Plate Growth Assays

Strains were grown overnight in YNB + 1% acetate + 50 µM BPS for 18 h at 30 °C with agitation. Strains were washed twice with autoclaved ddH_2_O, then diluted to an OD600 of 3 in ddH_2_O. Five-fold serial dilutions were spotted on appropriate media using a multichannel pipette. Plates were incubated at 37 °C for 96 h and imaged.

### 2.4. RNA-Seq

Our RNA-seq workflow was as described previously [10]. In brief, YNB + 1% sodium acetate overnight cultures were used to inoculate prewarmed 125 mL flasks with 25 mL YNB + 1% sodium citrate to an OD600 of 0.2. Cultures were grown for 4 h at 37 °C with agitation, and three biological replicates were used for sequencing. Cells were harvested via vacuum filtration, frozen (−80 °C), and RNA was extracted as described [10,20]. Novogene conducted mRNA sequencing and basic bioinformatic analyses with 1 μg RNA per sample. Sequencing reads were aligned to the *C. albicans* reference genome (Assembly 22) using Hisat2 v2.0.5. Differential expression analysis between 2 groups (3 biological replicates per group) was performed using the DESeq2 v1.40.2 R package using alpha = 0.05 [10,21].

### 2.5. Data Interpretation

Interpretations and hypotheses were always guided by the comprehensive information at the Candida Genome Database [22], FungiDB [23], and the KEGG database [24]. GO term enrichments were determined with the GO Termfinder tool at the Candida Genome Database.

### 2.6. Data Availability


Strains and plasmids are available upon request. The authors affirm that all data necessary for confirming the conclusions of the article are present within the article, figures, and tables. RNA-seq data have been deposited in the NCBI Gene Expression Omnibus with accession number GSE287330.

## 3. Results

### 3.1. Impact of HGT17 on the adr1Δ/Δ Mutant Growth Phenotype

*HGT17* is among the most highly Adr1-dependent genes. We reasoned that reduced *HGT17* expression may contribute to the *adr1*Δ/Δ growth defect on citrate. To test that idea, we asked whether increased *HGT17* expression may improve growth of an *adr1*Δ/Δ mutant on citrate medium. We fused the *HGT17* coding region to the iron-repressed *RBT5* promoter [14] in an *adr1*Δ/Δ mutant and tested its growth on media supplemented with bathophenanthrolinedisulfonate (BPS), an iron chelator that increases *RBT5* promoter activity. On glucose medium (Figure 1A), the three control strains (wild type, *adr1*Δ/Δ, and *hgt17*Δ/Δ) grew equally well, and two independent *adr1*Δ/Δ *RBT5-HGT17* strains grew comparably to the controls. On citrate medium (Figure 1B,C), the wild type grew much better than the *adr1*Δ/Δ and *hgt17*Δ/Δ mutants, as expected [10]. The two independent *adr1*Δ/Δ *RBT5-HGT17* strains grew better than the *adr1*Δ/Δ strain, though not as well as the wild type. Similar observations were made with 50 µM BPS (Figure 1B) or 100 µM BPS (Figure 1C). This result argues that *HGT17* is a key functional Adr1 target for growth on citrate, because increased *HGT17* expression can partially compensate for the absence of Adr1.

### 3.2. Hgt17-Dependent Gene Expression

Reduced *HGT17* expression may contribute not only to the *adr1*Δ/Δ growth defect but also to its regulatory impact. To explore that idea, we conducted RNA-seq analysis of the *hgt17*Δ/Δ mutant and wild-type strain on citrate medium. Under these conditions, 2676 genes had altered RNA levels (1360 downregulated, 1316 upregulated) in the *hgt17*Δ/Δ strain compared to the wild type (using conventional cutoffs of Log_2_ Fold Change [LFC] <−1 or >1 and an adjusted *p*-value < 0.05 [Appendix A). Upregulated genes were enriched for functions related to ribosome biogenesis; downregulated genes were enriched for diverse metabolic functions, including amino acid biosynthesis, ergosterol biosynthesis, and the citric acid cycle (Appendix A). These results indicate that Hgt17, like Adr1, has global impact on gene expression in citrate medium. The growth defect of the *hgt17*Δ/Δ mutant is likely to be a driver of its gene expression impact.

There was considerable overlap between the genes affected by the *hgt17*Δ/Δ mutation and those previously reported [10] for the *adr1*Δ/Δ mutation (Figure 2A). The functionally relevant citric acid cycle genes and glycolytic/gluconeogenic genes showed similar but not identical expression changes in response to each mutation (Figure 2B,C; R^2^ = 0.55−0.63 between LFC values). Many of the genes most strongly downregulated in the *hgt17*Δ/Δ and *adr1*Δ/Δ mutants were also strongly induced in citrate medium compared to acetate medium (Figure 2B,C). A correlation between impact of *hgt17*Δ/Δ and *adr1*Δ/Δ was also evident among ergosterol biosynthesis genes (Figure 2D; R^2^ = 0.55), many of which are direct Adr1 targets [9]. The similarity between *hgt17*Δ/Δ and *adr1*Δ/Δ gene expression changes stands in contrast to the considerable upregulation of *ADR1* RNA levels in the *hgt17*Δ/Δ mutant (Appendix A), a possible indication that Adr1 is inactive in citrate medium in the absence of Hgt17. We note that a few amino acid transporter genes (*GAP2*, *GAP5*, *GAP6*, *HIP1*, *MUP1*) were downregulated in the *hgt17*Δ/Δ mutant and not in the *adr1*Δ/Δ mutant. There could be something interesting with those transporters, so, overall, these results show that *adr1*Δ/Δ and *hgt17*Δ/Δ mutations have similar impact on a range of gene sets and argue that reduced *HGT17* expression contributes to the gene expression impact of an *adr1*Δ/Δ mutation.

## 4. Discussion

Our study here was driven by the finding that *HGT17* is among the most downregulated genes in an *adr1*Δ/Δ mutant compared to the wild type [10], prompting the idea that the *HGT17* expression defect may account for a portion of the *adr1*Δ/Δ mutant phenotype. While we acknowledge that the molecular function of Hgt17 is not entirely established, we interpret our results in view of the hypothesis that Hgt17 transports citrate.

Adr1 positively regulates numerous citric acid cycle and gluconeogenic genes [10]. How could it be then that overexpression of one target gene, *HGT17*, can improve growth on citrate? We interpret the result to indicate that the *adr1*Δ/Δ citrate growth defect is an aggregate effect of (i) diminished citrate uptake and (ii) diminished citrate metabolism capability. Overcoming either of these factors would thus improve citrate growth. However, both factors would have to be remedied to fully restore growth on citrate to wild-type levels. In fact, there is the suggestion in the literature that *HGT17* expression may be limiting even for *ADR1+/+* strains to grow on citrate: an *eed1*Δ/Δ mutation improves growth on citrate, and *HGT17* is among the most highly upregulated genes in that mutant [11].

Why would citrate uptake need to be regulated? The reason may be that citrate accumulation can inhibit glycolysis. In *S. cerevisiae*, metabolic flux analysis and biochemical assays show that internal citrate accumulation causes severe inhibition of pyruvate kinase (Cdc19) [25]. Therefore, internal citrate accumulation can impede growth on substrates that enter glycolysis upstream of phosphoenolpyruvate. If *C. albicans* citrate uptake were constitutive, then external citrate could enter the cell and inhibit pyruvate kinase, blocking use of hexoses and glycerol. Tight control of *HGT17* expression thus enables cells to consume preferred energy sources in the presence of citrate.

Hgt17 has a prominent effect on gene expression in citrate medium. For that reason, one might consider that Hgt17 has a direct role in citrate signaling. Glucose transporter homologs that function as signaling receptors include *C. albicans* Hgt4 and *S. cerevisiae* Snf3 and Rgt2 [26]. We consider such a role for Hgt17 unlikely for two reasons. First, the signaling receptors have a large C-terminal domain not shared with transporters [26], and Hgt17 lacks that domain. Second, the signaling receptors are expressed at low levels, an early clue that they could not transport sufficient substrate for growth [27]. However, *HGT17* has RNA levels comparable to bona fide transporter genes *HGT8* and *HGT7* and ~80-fold higher RNA levels than signaling receptor gene *HGT4* [10].

How then does Hgt17 have such strong impact on gene expression? We suggest two simple explanations. One possibility is that the gene expression changes represent the effects of carbon starvation. That is, citrate is the only carbon source in the medium, and if it cannot be taken up, then cells may use only endogenous carbon reserves. A second possibility, admittedly an entirely speculative one, is that citrate may promote transcriptional activation by Adr1. For example, citrate or a derived metabolite may be an allosteric effector that stimulates Adr1 binding to DNA. (This latter explanation is analogous to inducer exclusion in the *lac* operon system [28]). These explanations are not mutually exclusive, so both concepts may contribute to the *hgt17*Δ/Δ gene expression defect. We hope that future investigators will test some of the ideas we have generated through this now unfunded and unstaffed project.

## Figures and Tables

**Figure 1 jof-11-00889-f001:**
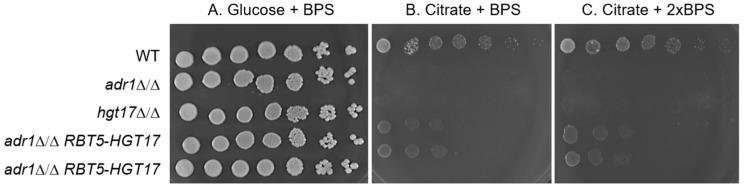
Growth properties of *adr1*Δ/Δ and *adr1Δ/Δ RBT5-HGT17* strains. The indicated strains were grown overnight in YNB plus 1% sodium acetate + 50 µM bathophenanthrolinedisulfonate (BPS) broth cultures, then five-fold serial dilutions were spotted on (**A**) YNB + 2% glucose + 50 µM BPS, (**B**) YNB + 1% sodium citrate + 50 µM BPS, and (**C**) YNB + 1% sodium citrate + 100 µM BPS. Plates were imaged after 96 h at 37 °C. We used acetate as the carbon source in overnight cultures because it is permissive for the wild type, the *adr1* mutant, and the *hgt17* mutant [10]. Strains used were MC22 (WT), AW10 (*adr1*Δ/Δ), AW157 (*hgt17*Δ/Δ), AW313 and AW315 (*adr1Δ/Δ RBT5-HGT17*).

**Figure 2 jof-11-00889-f002:**
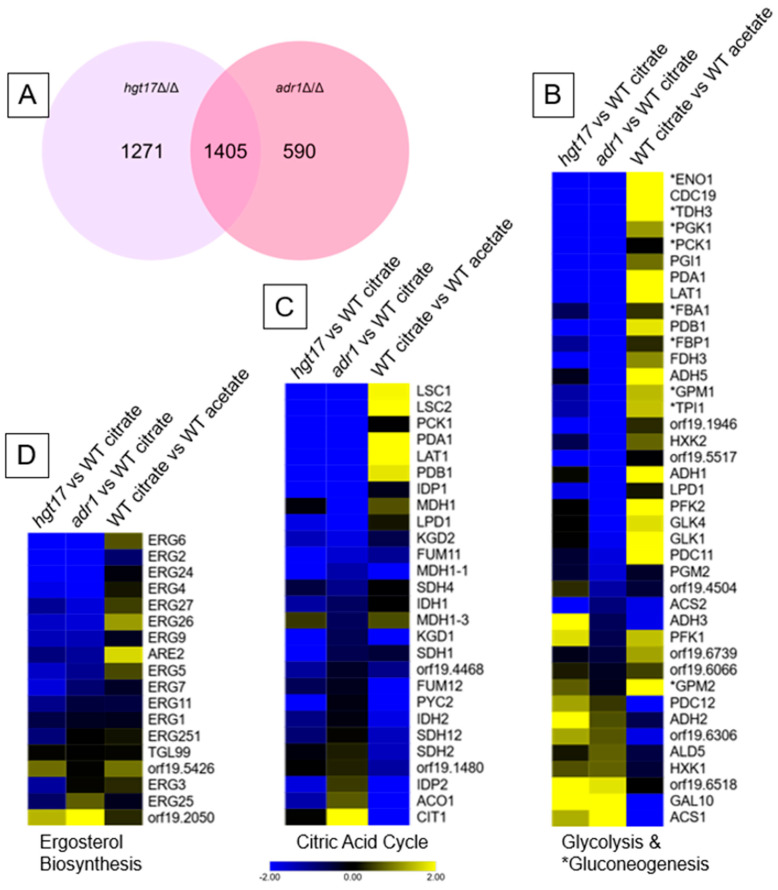
Gene expression changes in *hgt17*Δ/Δ vs. wild type comparison. (**A**) Venn diagram depicting significantly regulated genes (Log2Fold change <−1 or >1, padj < 0.05) in citrate-grown *hgt17*Δ/Δ vs. WT and *adr1*Δ/Δ vs. WT RNA-seq datasets. (**B**–**D**) Heat maps depicting LFC gene expression changes in selected biochemical pathways. (**B**) Glycolytic/Gluconeogenic genes; asterisks (*) denote Gluconeogenic genes. (**C**) Citric Acid Cycle genes. (**D**) Ergosterol Biosynthesis genes. A scale bar at the bottom shows the color scale for all three heatmaps. Datasets used are *hgt17*Δ/Δ vs. WT in citrate medium (this study), adr1Δ/Δ vs. WT in citrate medium [10], and WT in citrate medium vs. WT in acetate medium [10]. Gene sets were obtained from the KEGG database [24] entries at https://www.kegg.jp/entry/cal00100, https://www.kegg.jp/entry/cal00020, and https://www.kegg.jp/entry/cal00010, all accessed on 23 November, 2025.

## Data Availability

The data presented in this study are openly available in NCBI GEO at https://www.ncbi.nlm.nih.gov/geo/, reference number GSE287330, accessed on 16 December 2025.

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
