# Peer review of "Hgt17-Adr1 Relationship in Candida albicans Citrate Utilization"

_jof, 2025, doi:10.3390/jof11120889_

Round 1
Reviewer 1 Report
The authors should discuss why increased HGT17 dosage does not fully restore citrate growth to WT levels. This point should be explicitly addressed and emphasized in the Discussion section.
Lines 56–58: Replace “Overnights” with “overnight cultures.”
The sentence beginning with “For the experiment in Figure 1, …” should be rephrased for clarity.
Line 75: What is CSM ?
Author Response
1. The authors should discuss why increased HGT17 dosage does not fully restore citrate growth to WT levels. This point should be explicitly addressed and emphasized in the Discussion section.
Response: In the revised Discussion, lines 177-183, we now say: "Adr1 positively regulates numerous citric acid cycle and gluconeogenic genes (10). How could it be then that overexpression of one target gene, HGT17, can improve growth on citrate? We intepret the result to indicate that the adr1Δ/Δ citrate growth defect is an aggregate effect of (i) diminished citrate uptake and (ii) diminished citrate metabolism capability. Overcoming either of these factors would thus improve citrate growth. However, both factors would have to be remedied to fully restore growth on citrate to wild-type levels."
2. Lines 56–58: Replace “Overnights” with “overnight cultures.”
Response: I used a global search to correct all uses of this colloquial terminology.
3. The sentence beginning with “For the experiment in Figure 1, …” should be rephrased for clarity.
Response: We now say, "For the experiment using BPS in growth tests, overnight cultures included 50 µM BPS and plates included 50 or 100 µM BPS as indicated; BPS increases RBT5 promoter activity (14)."
4. Line 75: What is CSM?
Response: CSM is a conventional and convenient supplement mixture used to make complete or dropout media when added to YNB and dextrose. It is quite widely available, from Thermofisher and other suppliers. We modified the sentence in an effort to clarify: "All C. albicans transformants were selected as described (10) on Complete Supplement Mixture lacking histidine (CSM-HIS medium; 0.67% yeast nitrogen base without amino acids, 0.079% CSM-HIS supplement mixture, and 2% dextrose) for His+ isolates or "
Reviewer 2 Report
This study investigates the role of the transcription factor Adr1 and the gene HGT17 in enabling Candida albicans to utilize citrate as a carbon source. The authors propose that HGT17 functions as a citrate transporter and demonstrate that its expression is Adr1-dependent.
Major concerns
The central claim that HGT17 is a citrate transporter is based on indirect evidence (growth phenotypes and expression patterns). Would it be necessary to include direct transport assays (e.g., radiolabeled citrate uptake or biochemical validation) to confirm transporter activity?
Growth assays use 1% sodium citrate and iron chelator BPS to activate the RBT5 promoter. Are these conditions physiologically relevant to host environments? Could iron limitation or BPS independently affect citrate metabolism?
Overexpression partially rescues growth in adr1Δ/Δ mutants but does not restore wild-type levels. Would the authors provide quantitative growth curves or biomass measurements to complement spot assays and clarify whether other Adr1 targets are limiting.
The overlap between adr1Δ/Δ and hgt17Δ/Δ mutants is moderate (R² ≈ 0.55–0.63).
The discussion suggests citrate or its metabolites may act as Adr1 inducers. Either provide preliminary evidence (e.g., Adr1 binding assays in citrate vs acetate) or clearly state this as speculative.
This study investigates the role of the transcription factor Adr1 and the gene HGT17 in enabling Candida albicans to utilize citrate as a carbon source. The authors propose that HGT17 functions as a citrate transporter and demonstrate that its expression is Adr1-dependent.
Major concerns
The central claim that HGT17 is a citrate transporter is based on indirect evidence (growth phenotypes and expression patterns). Would it be necessary to include direct transport assays (e.g., radiolabeled citrate uptake or biochemical validation) to confirm transporter activity?
Growth assays use 1% sodium citrate and iron chelator BPS to activate the RBT5 promoter. Are these conditions physiologically relevant to host environments? Could iron limitation or BPS independently affect citrate metabolism?
Overexpression partially rescues growth in adr1Δ/Δ mutants but does not restore wild-type levels. Would the authors provide quantitative growth curves or biomass measurements to complement spot assays and clarify whether other Adr1 targets are limiting.
The overlap between adr1Δ/Δ and hgt17Δ/Δ mutants is moderate (R² ≈ 0.55–0.63).
The discussion suggests citrate or its metabolites may act as Adr1 inducers. Either provide preliminary evidence (e.g., Adr1 binding assays in citrate vs acetate) or clearly state this as speculative.
Author Response
1. Improve heatmap readability in Figure 2 (include gene names or IDs for key pathways).
Response: So sorry! Between the panel arrangement and pdf conversion the text was very difficult to read. We have now remade Figure 2. The panel arrangement ensures a slightly larger image, and the heatmaps were remade to improve gene name legibility. The three heatmaps present separately glycolytic/gluconeogenic gene, citric acid cycle genes, and ergosterol biosynthetic genes, using the pathway descriptions from the KEGG database.
2. The central claim that HGT17 is a citrate transporter is based on indirect evidence (growth phenotypes and expression patterns). Would it be necessary to include direct transport assays (e.g., radiolabeled citrate uptake or biochemical validation) to confirm transporter activity?
Response: Although we previously proposed that Hgt17 is a citrate transporter (ref 10), we do not conclude that it is here. In fact, we routinely qualified that functional assignment in this manuscript for the very reason you noted - that evidence is indirect. Our study here was motivated by the observation that HGT17 is more highly dependent on Adr1 than most genes. I agree that proof that HGT17 is a citrate transporter would require uptake assays, but we do not seek to prove that point. I deleted most of the first paragraph of the Discussion to ensure that no reader would think that we have proved that Hgt17 is a citrate transporter.
3. Growth assays use 1% sodium citrate and iron chelator BPS to activate the RBT5 promoter. Are these conditions physiologically relevant to host environments? Could iron limitation or BPS independently affect citrate metabolism?
Response: We do not claim that the growth conditions are relevant to the host environment. Based on the human metabolome database, there is probably no host environment in which citrate is the sole carbon source. So, in order to analyze citrate utilization, we had to take a reductionist approach and use artificial growth conditions. Our study may be viewed as a basic biological study, not a pathogenesis study. Regarding the impact of BPS and iron limitation on citrate metabolism, we expect severe iron limitation to block citrate utilization due to disruption of the electron transport chain. For this reason we used the lowest concentration of BPS needed to induce the RBT5 promoter (ref (14)), about 8-fold less than we typically use.
4. Overexpression partially rescues growth in adr1Δ/Δ mutants but does not restore wild-type levels. Would the authors provide quantitative growth curves or biomass measurements to complement spot assays and clarify whether other Adr1 targets are limiting.
Response: I regret that I do not have personnel or funds to do more experiments on this subject. I lightened all images in Figure 1 to hopefully make the phenotypic rescue obvious. We hope that future investigators will pick up where we left off.
5. The overlap between adr1Δ/Δ and hgt17Δ/Δ mutants is moderate (R² ≈ 0.55–0.63).
Response: Agreed. As noted in the last paragraph of the Discussion, there are probably two drivers of the hgt17 phenotype - carbon starvation and Adr1 inactivity.
6. The discussion suggests citrate or its metabolites may act as Adr1 inducers. Either provide preliminary evidence (e.g., Adr1 binding assays in citrate vs acetate) or clearly state this as speculative.
Response: We modified the sentence to emphasize that this model is speculative. "A second possibility, admittedly an entirely speculative one, is that citrate may promote transcriptional activation by Adr1."
Reviewer 3 Report
This manuscript contains the only two data; 1) growth suppression phenotype of the Candida adr1/adr1 deletion mutant by expression of HGT17 on the citrate medium; 2) RNA seq analysis of hgt1/hgt17 deletion mutant. No deep analysis was conducted.
They should try more experiments to support their conclusion, such as time course gene expression profile of HGT17 in WT and adr1/adr1 deletion mutant and a proof of the direct binding of Adr1 with HGT17 by ChIP.
Author Response
1. Figure1, spots of three pictures do not align. I suspect the data was taken separately. It has to be taken at the same time.
Response: Thanks so much for this insightful comment. Actually the images were taken at the same time, just as you said they should be.
2. Page5, the authors put Figure 4, but only Figure 2 is presented.
Response: Apologies! This error has been fixed in the revision.
3. This manuscript contains the only two data; 1) growth suppression phenotype of the Candida adr1/adr1 deletion mutant by expression of HGT17 on the citrate medium; 2) RNA seq analysis of hgt1/hgt17 deletion mutant. No deep analysis was conducted.
Response: We agree with your summary. This is a simple study with simple conclusions.
4. They should try more experiments to support their conclusion, such as time course gene expression profile of HGT17 in WT and adr1/adr1 deletion mutant and a proof of the direct binding of Adr1 with HGT17 by ChIP.
Response: These are excellent suggestions that we hope future investigators will pursue - maybe even you! However, given the lack of funding and personnel on the project right now, we are unable to carry out these experiments.
Reviewer 4 Report
In this manuscript, White and Mitchell present a study regarding the Hgt17-Adr1 relationship in the pathogen Candida albicans under citrate growth conditions. The authors argue that HGT17 expression impacts growth and gene expression when citrate is the only carbon source, especially in an adr1Δ/Δ mutant.
The study is timely and of relevance for understanding the metabolic flexibility of pathogen fungi coordinated by nutrient sensors/transporters and transcriptional regulatory networks.
There are some issues that must be addressed before the manuscript can be considered for publication.
- Figure 1 is not sufficiently clear. Specifically, it is not evident that HGT17 overexpression improves the growth properties of adr1Δ/Δ. Also, it would be interesting to know if HGT17 overexpression restores the growth on citrate-containing media of hgt17Δ/Δ cells or even augments the growth of wild type cells under the same conditions.
- Line 56: please define YNB medium and VWR.
- Line 57: BPS must be explained. Although this is done in Figure 1 legend, the abbreviation must also be defined in the text.
N/A
Author Response
1. The images in Figure 1 are not clear. Figures 1B and 1C appear redundant.
Response: I brightened up all of the images equivalently and I think they are more clear now. See what you think. As for 1B and 1C redundancy, since we are using fairly low levels of BPS compared with ref 14, we wanted to see if increased levels would improve the response. We are walking a bit of a tightrope because high levels of BPS inhibit electron transport and hence citrate utilization.
2. Figure 1 is not sufficiently clear. Specifically, it is not evident that HGT17 overexpression improves the growth properties of adr1Δ/Δ. Also, it would be interesting to know if HGT17 overexpression restores the growth on citrate-containing media of hgt17Δ/Δ cells or even augments the growth of wild type cells under the same conditions.
Response: I brightened up all of the images equivalently and I think they are more clear now. See what you think. The controls that you describe are great ideas, but we do not have those results.
3. Line 56: please define YNB medium and VWR.
YNB is yeast nitrogen base without amino acids. VWR is the scientific supply company from which we purchased sodium acetate. It is quite well known. To clarify these points I modified the sentence: "Overnight cultures were grown in filter sterilized yeast nitrogen base without amino acids + 1% sodium acetate (supplied by VWR) liquid medium." Is that adequate?
4. Line 57: BPS must be explained. Although this is done in Figure 1 legend, the abbreviation must also be defined in the text.
Response: It is described on line 121 in the Results section, as it was in Results on line 133 in the original submission. It is possible that I misunderstand the comment, for which I apologize.
Reviewer 5 Report
The manuscript by White et al. explores the relationship between Adr1 and Hgt17 in citrate utilization by C. albicans. The study is well performed overall, and the results contribute to understanding carbon metabolism in C. albicans. However, the manuscript would benefit from some scientific and technical clarifications, as outlined below.
- It would be helpful if the authors could provide a concise quantification of HGT17 expression under the RBT5 promoter (± BPS)? This could help readers assess the extent of overexpression relative to the wild type.
- The overlap between the two datasets is interesting. Would the authors consider adding a short statement indicating whether any major genes or pathways differ substantially between the two mutants?
- The images in Figure 1 appear to have low contrast; please improve their resolution. Additionally, provide information on the biological replicates and indicate whether the presented images are representative of three or more independent experiments.
- Provide rationale for using acetate in overnight cultures.
- The figure labeling in the result section appears to be erroneous; it should refer to Figure 2 rather than Figure 4.
The manuscript by White et al. explores the relationship between Adr1 and Hgt17 in citrate utilization by C. albicans. The study is well performed overall, and the results contribute to understanding carbon metabolism in C. albicans. However, the manuscript would benefit from some scientific and technical clarifications, as outlined below.
- It would be helpful if the authors could provide a concise quantification of HGT17 expression under the RBT5 promoter (± BPS)? This could help readers assess the extent of overexpression relative to the wild type.
- The overlap between the two datasets is interesting. Would the authors consider adding a short statement indicating whether any major genes or pathways differ substantially between the two mutants?
- The images in Figure 1 appear to have low contrast; please improve their resolution. Additionally, provide information on the biological replicates and indicate whether the presented images are representative of three or more independent experiments.
- Provide rationale for using acetate in overnight cultures.
- The figure labeling in the result section appears to be erroneous; it should refer to Figure 2 rather than Figure 4.
Author Response
The manuscript by White et al. explores the relationship between Adr1 and Hgt17 in citrate utilization by C. albicans. The study is well performed overall, and the results contribute to understanding carbon metabolism in C. albicans. However, the manuscript would benefit from some scientific and technical clarifications, as outlined below.
- It would be helpful if the authors could provide a concise quantification of HGT17 expression under the RBT5 promoter (± BPS)? This could help readers assess the extent of overexpression relative to the wild type.
- Response: I regret to say that we did not conduct this analysis. We have found that RBT5 promoter fusions to different genes can vary a bit in terms of RNA levels, so we prefer not to speculate.
- The overlap between the two datasets is interesting. Would the authors consider adding a short statement indicating whether any major genes or pathways differ substantially between the two mutants?
- Response: Thank you for this excellent suggestion! Regrettably though there is not much to say. I used the CGD termfinder tool to analyze genes upregulated in the hgt17 mutant, and not in the adr1 mutant; there was no significant enrichment, probably a reflection of the fact that more than half the genes were assigned to "biological process unknown." For genes downregulated in the hgt17 mutant, and not in the adr1 mutant; there was a very modest enrichment (p = 0.00061) for amino acid transporters, including GAP2, GAP5, GAP6, HIP1, MUP1. There could be something interesting with those transporters, so I added this sentence at line 163, "We note that a few amino acid transporter genes (GAP2, GAP5, GAP6, HIP1, MUP1) were downregulated in the hgt17Δ/Δ mutant and not in the adr1Δ/Δ mutant."
- The images in Figure 1 appear to have low contrast; please improve their resolution. Additionally, provide information on the biological replicates and indicate whether the presented images are representative of three or more independent experiments.
- Response: The contrast has been improved in the revision; see what you think. The data we present are from two independent adr1Δ/Δ RBT5-HGT17 strains.
- Provide rationale for using acetate in overnight cultures.
- Response: We use acetate in overnights because it is permissive for the wild type, the adr1 mutant, and the hgt17 mutant. We have added this statement to the Figure 1 legend: "We used acetate as the carbon source in overnight cultures because it is permissive for the wild type, the adr1 mutant, and the hgt17 mutant (10)."
- The figure labeling in the result section appears to be erroneous; it should refer to Figure 2 rather than Figure 4.
- Response: Apologies for the error. This has been corrected in the revision.
Round 2
Reviewer 2 Report
The authors have addressed all of my concerns with the original manuscript
The authors have addressed all of my concerns with the original manuscript
Author Response
Many thanks for your interest in our work!
Reviewer 3 Report
I am sorry to tell that the authors response is unsatisfactory.
I do not know the journal will accept this preliminary work.
Reviewer 4 Report
Dear Authors,
Please address the following:
Lines 56-57: describe the synthetic medium used: i. e., percentage of YNB, pH, and whether you added anything else (such as mixture of amino acids?) Was the medium selective for plasmid maintanance?
Also, for VWR supplier, specify town and country.
N/A
Author Response
1. Lines 56-57: describe the synthetic medium used: i. e., percentage of YNB, pH, and whether you added anything else (such as mixture of amino acids?) Was the medium selective for plasmid maintanance?
Response: Apologies for the omission. We have now indicated that the medium contained 0.67% yeast nitrogen base without amino acids (along with acetate). We did not adjust the pH, so this point is not mentioned. No amino acids were added, so that is not mentioned. The cells did not contain a plasmids; instead the RBT5 promoter was integrated into the genome upstream of one of the HGT17 coding regions.
2. Also, for VWR supplier, specify town and country.
Response: Our purchases are made through a state contractor (called UGAmart), so we do not know the actual town and country from which VWR shipped. VWR headquarters are in Radnor, PA, USA. Did you want us to add that information?